# Learning in Order!
# A Sequential Strategy to Learn Invariant Features for Multimodal Sentiment Analysis

Submission Id 2340

## ABSTRACT

This work proposes a novel and simple sequential learning strategy to train models on videos and texts for multimodal sentiment analysis. To estimate sentiment polarities on unseen out-of-distribution data, we introduce a multimodal model that is trained either in a single source domain or multiple source domains using our learning strategy. This strategy starts with learning domain invariant features from text, followed by learning sparse domain-agnostic features from videos, assisted by the selected features learned in text. Our experimental results demonstrate that our model achieves significantly better performance than the state-of-the-art approaches on average in both single-source and multi-source settings. Our feature selection procedure favors the features that are independent to each other and are strongly correlated with their polarity labels. To facilitate research on this topic, the source code of this work will be publicly available upon acceptance.

## KEYWORDS

MSA, OOD, Invariant Features

## 1 INTRODUCTION

Multimodal Sentiment Analysis (MSA) is concerned with understanding people's attitudes or opinions based on information from more than one modalities, such as videos and texts. It finds rich applications in both industry and research communities, such as understanding spoken reviews of target products posted on YouTube and developing multimodal AI assistants for mental health support. Prior MSA approaches make an impractical assumption that training and test data comprise independent identically distributed samples [58, 64, 76, 79–81]. However, training datasets are available only for a handful of applications that satisfy that assumption. Therefore, this work aims to remove the assumption such that MSA models trained on a single domain or multiple source domains can work robustly on *unseen* out-of-distribution (OOD) data, without leveraging any target domain data.

To enable models to work robustly across domains, a key idea is to exploit domain invariant sparse representations, which serve as causes of target labels from a causal perspective [63, 67]. In contrast, spurious correlations, which do not indicate causal relations, impede the generalization capability of pre-trained foundation models [6, 25]. Existing MSA models heavily rely on jointly learned multimodal features for sentiment analysis [27]. However, the spurious features of the visual modality may adversely affect the features of the text modality, leading to inaccurate prediction outcomes [23, 24, 80]. Therefore, it would be interesting to investigate **i) how to automatically identify domain invariant representations**

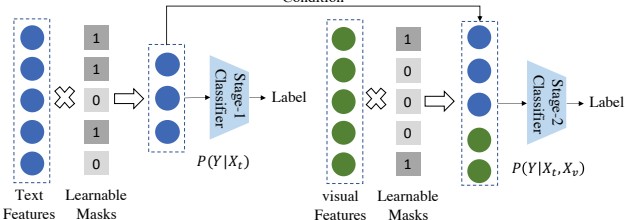

**Figure 1: Classifiers employ learnable masks to identify domain-invariant text features first, conditioned on which the classifiers learn domain-invariant features from videos.**

**for MSA, and ii) what are the key characteristics of domain invariant features in a multimodal setting**.

To answer the above research questions, as illustrated in Figure 1, we propose a **S**equential **S**trategy to **L**earn **I**nvariant **F**eatures (S$^2$LIF) for building a domain generalization (DG) MSA model based on videos and texts. Instead of learning domain-agnostic features simultaneously from all modalities, our technique first leverages the sparse masking technique [34] to select invariant hidden features from texts, followed by learning the invariant features from videos, conditioned on the selected textual features. *To the best of our knowledge, it is the first time to report the importance of feature learning order for domain generalization.* We conduct extensive experiments to i) demonstrate the superiority of our approach in comparison with the competitive baselines in both single source domain and multi-source domain settings, and ii) investigate key characteristics of selected features using our approach. Our key contributions are summarized as follows:

- We introduce a novel domain generalization MSA model, which explicitly learns domain-invariant features and mitigates spurious domain-specific features by adopting a sparse masking technique.
- We propose a new sequential multimodal learning strategy, which extracts the domain-invariant features from texts first, followed by employing them to identify the features relevant to labels from videos using the sparsity technique.
- We demonstrate empirically that i) our sequential multimodal learning strategy prefer selecting the domain invariant features in the visual modality, which are independent of the selected features in the text modality and strongly correlate with labels; and ii) it is important to adhere to the learning order of our approach to mitigate spurious correlations, because it is evident that the alternative learning

order or learning all modalities simultaneously using the same sparsity technique leads to inferior performance.

## 2 RELATED WORK

### 2.1 Multimodal Sentiment Analysis

MSA methods can be roughly divided into two categories: 1) Multimodal Representation Learning aims to learn fine-grained multimodal representation, which provides rich decision evidence for multimodal sentiment prediction. They employ a disentangled technique to learn modality-common and modality-specific representations to mitigate the heterogeneity of multimodal representations [28, 65, 71, 74]. 2) Multimodal Fusion aims to learn cross-modal information transfer by designing complex cross-modal interactive networks. The development of multimodal fusion methods has evolved from multi-modal tensor fusion [76] to cross-modal attention[44, 45, 64, 73, 77, 79–82]. The current MSA methods only train and test on a specific domain, and do not consider the generalization ability of the model. They suffer performance degradation when tests on out-of-distribution data, so learning robust MSA models is essential.

### 2.2 Domain Generalization

Domain generalization aims to design a deep neural network model that learns domain-invariant features and is able to maintain stable performance in both the source domain and multiple unseen target domains. Numerous domain generalization methods have been proposed to learn domain-invariant features for single-source or multi-source domain generalization[5, 12–18, 22, 26, 30, 31, 37, 38, 40, 48, 51, 54, 59, 72]. We roughly divide current methods of domain generalization into three categories: 1) Learning invariant features aims to capture the domain-generalized features to reduce the dependence of features on specific domains and to achieve high performance on unseen domains[48]. 2) Optimize algorithm aims to learn domain-invariant features and remove domain-specific features[5, 10, 22, 38, 52, 59], such as adversarial training and meta-learning, through tailored designed network structures. 3) Data augmentation aims to generate new data to improve the generalization performance of the model, and these generated new data are out-of-distribution samples different from the source domain[66, 68, 69].

### 2.3 Causal Representation Learning

From the perspective of data generation, causal representation learning considers that raw data is entangled with two parts of features: correlated features with label (domain-invariant features) and spuriously correlated features with the label (domain-specific features). The goal is to disentangle domain-invariant features and domain-specific features. Domain-invariant features guarantee stable performance in different test environments[1, 49]. Based on this assumption, numerous methods attempt to learn domain invariant features [2, 3, 11, 32, 56]. Following previous work, our proposed approach aims to learn the domain-invariant features (i.e., the features correlated with the label), while removing the features domain-specific (i.e., the spuriously correlated features with the label). Concretely, we adopt sparse techniques to remove spuriously correlated features with the label [36, 43, 47, 60].

## 3 METHOD

**Problem Statement.** The goal of domain generalization for MSA is to train a deep neural network model on a single-source or multi-source domains $\mathcal{D}_S = \{\mathcal{D}_S^1, \mathcal{D}_S^2, ..., \mathcal{D}_S^N\}$ and evaluate the model on the unseen target domains $\{\mathcal{D}_T^1, \mathcal{D}_T^2, ..., \mathcal{D}_T^M\}$, where $\mathcal{D}$ denotes a dataset in a domain, $M$ and $N$ denote the number of source domains and target domains, respectively. We consider each MSA task as a k-ways classification task. The dataset in a source domain is denoted by $\mathcal{D}_S = \{\mathcal{X}_{\{t,v\}}^i, y_i\}_{i=1}^n$, where $\mathcal{X}_{\{t,v\}}^i \in \mathbb{R}^d$, $y \in \mathbb{R}^K$, while $\mathcal{D}_T$ is a dataset in a target domain. The goal is to learn multimodal domain-invariant features for sentiment polarity prediction in unseen domains without using target domain data for training.

**Model Overview.** Our work is motivated by the functional lottery ticket hypothesis [41] suggesting that there is a subnetwork that can achieve better out-of-distribution performance than the original network. Hence, We employ the sparse masking techniques to identify a subset of hidden features in the multimodal setting. The findings of our empirical studies indicate the importance of the learning order between modalities for domain generalization performance.

The architecture of our model is illustrated in Figure 2. Given a text and a sequence of video frames $\mathcal{X}_{\{t,v\}}$, we employ a pre-trained encoders ELECTRA [19] and VGGFace2 with a 1-layer Transformer encoder [21] to map them to respective hidden representations $x_t$ and $x_v$. To achieve sparsity in hidden representations, our model generates a mask vector $m_{\{t,v\}}$ with the mask function $f_{mask}$ to select domain-invariant features $x_{\{t,v\}}^c$ from $\mathcal{X}_{\{t,v\}}$. The mask function is characterized by the learnable parameter $r_{\{t,v\}}$ and threshold $s_{\{t,v\}}$. The feature selection in a modality is achieved by computing the dot-product between the mask vectors and the corresponding hidden representations. We empirically find that text is the superior modality in comparison with videos based on their performance in each modality. Our further studies show that conditioning on the strong text features reduces the selection of visual features that correlate with those text features. On the one hand, reduction of statistical dependencies between features leads to improvement of generalization performance. On the other hand, selection of features adhere to the functional lottery ticket hypothesis. Therefore, our text classifier $g_t$ first selects the key features using the masking technique, followed by learning sparse video representations for the visual classifier $g_v$ to predict sentiment polarity conditioned on the selected text features. In addition, our model leverages prior information from video frames to eliminate redundant frames.

**Keyframe-aware Masking.** Given that there is a large amount of frames in a video clip, which contains redundant information [47, 60]. The frame sequence $\bar{x}_v$ of a video clip contains rich priors, which explicitly correspond to neighboring frames. We can easily obtain the motion of the video frame sequence to guide the masking of redundant frames according to the temporal difference. We employ global and local neighbor frames to select informative frames $x_v$ and constrain the semantic invariance of video frames by reconstructing losses $\mathcal{L}_{recon}$. Note that this part is not our main contribution, and it is an extension of previous work [47]. See supplementary materials for details.

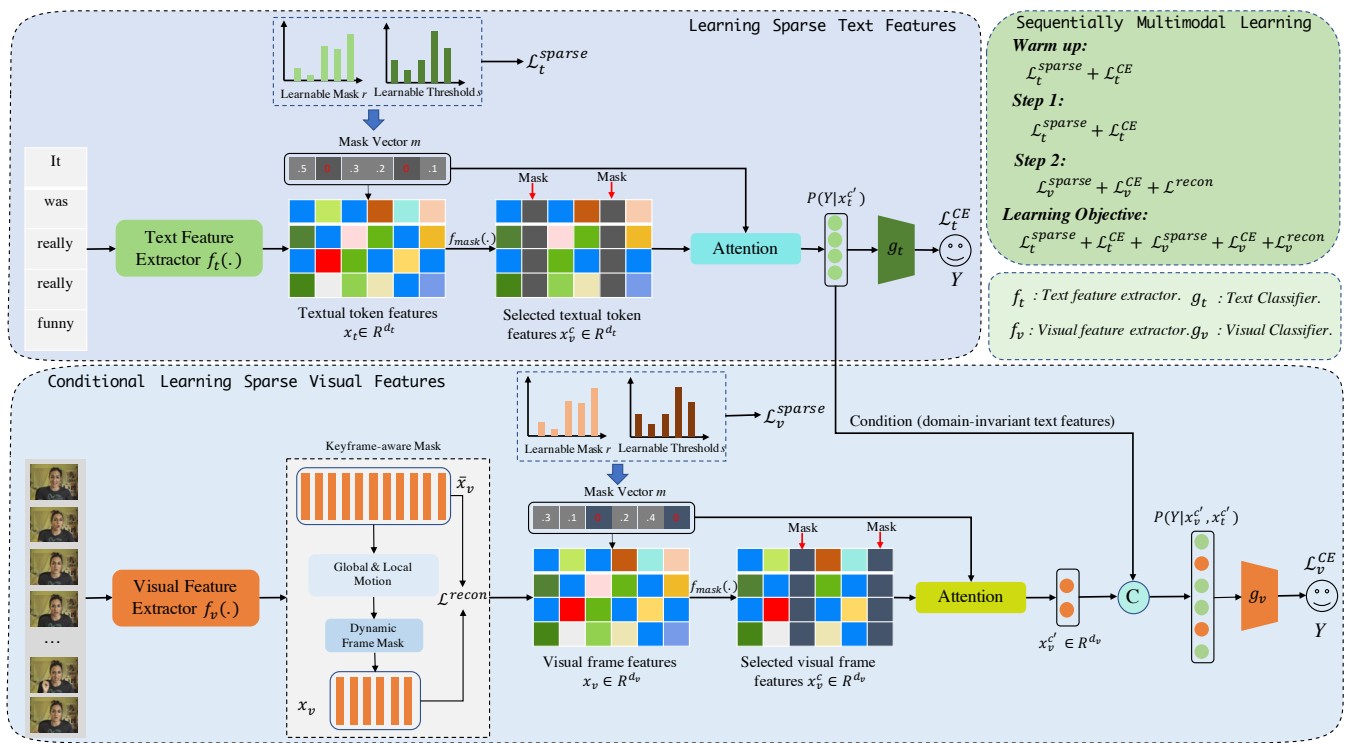

**Figure 2: An overview of our proposed framework.**

***Sequential Multimodal Learning.*** The selection of domain invariant features is also motivated from a causal perspective. The logit of the classifier is computed as the product between the features $x$ and the weights $W$ of the label $k$ from the classification layer $g$.

$$O_k = W_{\{,k\}}^T \cdot x = \sum_{* \in \{t,v\}} \sum_{j=1}^{d_t} W_{*_{\{j,k\}}} \cdot x_{*_j}, * \in \{t,v\}, \quad (1)$$

where the subscripts $j$ and $k$ denote $j$-th feature and $k$-th class respectively. For all polarity labels with both modalities, we obtain a matrix $R$ as follows, where each element $w_{*_{\{j,k\}}} \cdot x_{*_j}, * \in \{t,v\}$ represents the evidence of the classifier.

$$R = \begin{bmatrix} w_{t_{\{1,1\}}} x_{t_1} & \cdots & w_{t_{\{1,K\}}} x_{t_1} \\ w_{t_{\{2,1\}}} x_{t_2} & \cdots & w_{t_{\{2,K\}}} x_{t_2} \\ \vdots & \vdots & \ddots & \vdots \\ w_{t_{\{d_t,1\}}} x_{t_{d_t}} & \cdots & w_{t_{\{d_t,K\}}} x_{t_{d_t}} \\ w_{t_{\{1,1\}}} x_{v_1} & \cdots & w_{v_{\{1,K\}}} x_{v_1} \\ w_{t_{\{2,1\}}} x_{v_2} & \cdots & w_{v_{\{2,K\}}} x_{v_2} \\ \vdots & \vdots & \ddots & \vdots \\ w_{v_{\{d_v,1\}}} x_{v_{d_v}} & \cdots & w_{v_{\{d_v,K\}}} x_{v_{d_v}} \end{bmatrix} \quad (2)$$

By analyzing the matrix $R$, we conclude that **1)** $Y$ **is the result of feature** $x$ **estimated via a classifier.** From the causal perspective, the selected features can be seen as the causes of $Y$ subjecting to

independent noise [29, 50, 62]:

$$Y = g(Pa(Y)) + \epsilon \quad (3)$$

where the notation $Pa(Y)$ denotes the features of direct causal effects with Y, where $Pa(Y)$ is a subset of $x$. The function $g$ represents the classifier. The multimodal features $x$ are divided into two subsets, domain-specific features $x^s$ (spurious correlated features with the label across domain) and domain-invariant features $x^c$ (correlated features with the label across domain) [55]. We use three features $\{x_1, x_2, x_3\}$ to explain the causal relationship between $x$ and $Y$. As shown in Figure 3 (a), the outcome $Y$ is specified as $Y = g(x_1, x_3) + \epsilon, \{x_1, x_3\} \subseteq x^c$. The feature $x_3$ is the subset of $x^s$. There exist two distinct relationships between the feature sets $x^s$ and $x^c$: a) there is no direct causal relationship between $x_3$ and $x_1$. b) there is a direct causal relationship between $x_3$ and $x_2$. We remove the edge between $x_2$ and $x_3$ to eliminate the impact of $x_3$ on $x_2$. Therefore, our goal is to identify the features $x^c$ and remove the features $x^s$. Formally, we expect

$$\mathbb{P}(Y|do(x_i^c, x_k^s)) \neq \mathbb{P}(Y|do(x_j^c, x_k^s)) \quad (4)$$

where the features $\{x_i^c, x_j^c\} \subseteq x^c$ are selected mutual independent domain-invariant features [57]. We design learnable masks $m$ and learnable threshold $s$ in Section 3 to set the values of domain-specific features in $x_k^s$ to 0. Removing the features $x_k^s \subseteq x^s$ eliminates its direct causal effects on $(x_i^c, x_j^c)$ and the outcome $Y$. **2) simultaneously optimizing such entangled features** $x = \{x^c, x^s\}$ **for both text and visual modalities (i.e., imbalanced multimodal features) poses a special challenge for the classifier** [23, 24].

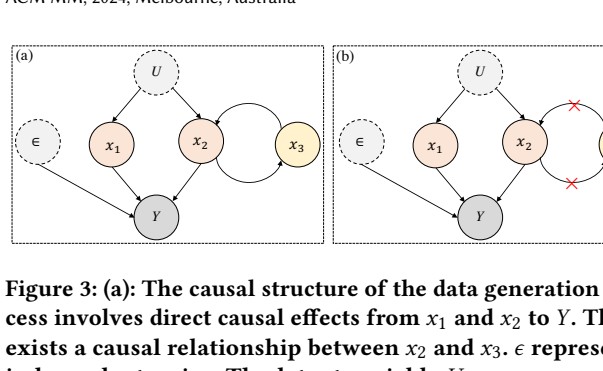

**Figure 3: (a): The causal structure of the data generation process involves direct causal effects from $x_1$ and $x_2$ to $Y$. There exists a causal relationship between $x_2$ and $x_3$. $\epsilon$ represents independent noise. The latent variable $U$ serves as a confounder for $x_1$ and $x_3$. (b) Severing the edge between $x_2$ and $x_3$ and eliminating the causal relationship.**

Our sequential learning strategy is also motivated by curriculum learning [8, 46, 84] that we learn the features first, which perform well on the target tasks, followed by more challenging ones.

By analyzing the causal relationship and removing spurious correlation features using multimodal learnable masks, we can obtain a new evidence matrix $R^M$. The form of the new evidence matrix $R^M$ for the classifier is as follows:

$$\mathbf{R}^M = \begin{bmatrix} w_{t_{\{1,1\}}} x_{t_1} m_{t_1} & \cdots & w_{t_{\{1,K\}}} x_{t_1} m_{t_1} \\ w_{t_{\{2,1\}}} x_{t_2} m_{t_2} & \cdots & w_{t_{\{2,K\}}} x_{t_2} m_{t_2} \\ w_{t_{\{3,1\}}} x_{t_3} m_{t_i} & \cdots & w_{t_{\{3,K\}}} x_{t_3} m_{t_i} \\ \vdots & \vdots & \ddots & \vdots \\ w_{t_{\{d_t,1\}}} x_{t_{d_t}} m_{t_{d_t}} & \cdots & w_{t_{\{d_v,K\}}} x_{t_{d_v}} m_{t_{d_v}} \\ w_{t_{\{1,1\}}} x_{v_1} m_{v_1} & \cdots & w_{v_{\{1,K\}}} x_{v_1} m_{v_1} \\ w_{t_{\{2,1\}}} x_{v_2} m_{v_2} & \cdots & w_{v_{\{2,K\}}} x_{v_2} m_{v_2} \\ w_{t_{\{3,1\}}} x_{v_3} m_{v_j} & \cdots & w_{v_{\{3,K\}}} x_{v_3} m_{v_j} \\ \vdots & \vdots & \ddots & \vdots \\ w_{v_{\{d_t,1\}}} x_{v_{d_t}} m_{v_{d_t}} & \cdots & w_{v_{\{d_v,K\}}} x_{v_{d_v}} m_{v_{d_v}} \end{bmatrix} \quad (5)$$

where $\{m_t, m_v\}$ denotes mask vector in Section 3. The red notation $m_{t_i}$ and $m_{v_j}$ represent learnable mask to select domain-invariant feature with two stages in the above equations. By analyzing the evidence matrix $R^M$ of the classifier $g$ and the direct causal effect with outcome $Y$, we can utilize the learnable mask and threshold to sequential select domain-invariant features $x^c$ and remove domain-specific features $x^s$.

***Multimodal Learnable Masks.*** Regarding how to automatically identify domain invariant representations for MSA, we design multimodal learnable masks to select features. Specifically, to remove domain-specific features, we tailor a function, denoted as $f_{mask}$. The inputs of $f_{mask}$ consists of the features $x$ from a modality, a learnable parameter $r$ , and a dynamic threshold $s$. The output is domain-invariant features $x^c$.

$$x^c = f_{mask}(x, r, s) \quad (6)$$

where we apply the mask vector $m \in \mathbb{R}^d$ (consisting of zero and non-zero value) on the feature $x \in \mathbb{R}^d$. The mask vector $m$ is obtained by utilizing a trainable pruning threshold $s \in \mathbb{R}^d$ and a learnable parameter $r \in \mathbb{R}^d$. Given a set of features $x$, our method can dynamically select features using mask vector $m$. We utilize the unit step function $\mathcal{F}(\cdot)$ to produce mask vector, which takes

the learnable parameters $r$ and thresholds $s$ as input and output the binary masks $p$. Formally,

$$\mathcal{F}(t) = \begin{cases} 0 & \text{if } t < 0 \\ 1 & \text{if } t \geq 0 \end{cases}, \quad (7)$$

where the binary mask $p$ and mask vector $m$ are obtained by:

$$p = \mathcal{F}(|r| - s), \quad (8)$$
$$m = r \odot p \quad (9)$$
$$x^c = x \odot m, \quad (10)$$

where $x^c$ represents domain-invariant features, which remove spuriously correlated features and retain the correlated features with the label $y$ in training stage. It was unable to complete end-to-end training during model training. The reason is that the binary mask produced by our unit step function is non-differentiable. To overcome this issue, previous works [33, 61, 83] based on straight-through estimator (STE) [7] to estimate derivatives and design binarization function that can be back-propagatation. [70] give more approximate estimates than STE to handle non-differentiable scenarios. Using this derivative estimate to approximate the unit step function allows the model to train end-to-end.

$$\frac{d}{dt}\mathcal{F}(t) = \begin{cases} 2 - 4|t|, & -0.4 \leq t \leq 0.4 \\ 0.4, & 0.4 \leq |t| \leq 1 \\ 0, & \text{otherwise} \end{cases} \quad (11)$$

To encourage the model to learn sparse features, we add a sparse regularization term [34] to the threshold as one of the training objectives. Formally,

$$\mathcal{L}_*^{sparse} = \sum_{i=1}^{N} exp(-s_i^*), * \in \{t, v\}, \quad (12)$$

where the regular term $exp(-s_i^*)$ raises the value of the dynamic threshold $s$, so that a few feature values can exceed the threshold to learn more sparse features. We utilize the function $f_{mask}$ to obtain domain-invariant features of textual and visual tokens. Formally,

$$x_*^c = f_{mask}(x_*, r_*, s_*), * \in \{t, v\} \quad (13)$$

where the definitions of $f_{mask}$, $r_*$, and $s_*$ are specified in Equation $6 \sim 10$.

Apart from learning the domain-invariant features of each token, we also calculate the similarity between each token and the learnable mask to learn domain-invariant tokens. Formally,

$$a*^{c_j} = sim(m_*, x_*^{c_j}), * \in \{t, v\}, \quad (14)$$

$$x_*^{c'} = \sum_{j=1}^{\tau_*} x_*^{c_j T} \cdot a*^{c_j}, * \in \{t, v\}, \quad (15)$$

where $x_*^{c_j}$ denotes $j$-th token of text and visual modalities. The $x_t^{c'}$ and $x_v^{c'}$ represent the features of fused domain-invariant tokens. The symbol $sim$ denotes similarity. The symbol $a$ denotes attention weight. The classifier $g_t$ and $g_v$ takes inputs $x_t^{c'}$ and $x_v^{c'}$, and outputs $logits\ O_t$ and $logits\ O_v$. Formally,

$$O_t = g_t(x_t^{c'}); O_v = g_v([x_v^{c'}; x_t^{c'}]) \quad (16)$$

where ';' denotes concatenation along the feature dimension.

*Learning Objective.* Initially, we employ the classifier to learn domain-invariant features from the text modality (i.e. text modality). Formally,

$$\mathcal{L}_t = \mathcal{L}_t^{CE} + \alpha \cdot \mathcal{L}_t^{sparse} \tag{17}$$

Subsequently, we utilize the domain-invariant features from the text modality to assist in selecting domain-invariant features from the visual modality. Formally,

$$\mathcal{L}_v = \mathcal{L}_v^{CE} + \alpha \cdot \mathcal{L}_v^{sparse} + \mathcal{L}_{recon} \tag{18}$$

where the symbol $\mathcal{L}_*^{CE}$ denotes *Cross-Entropy* loss and $\alpha$ is hyper-parameter. Accordingly, the overall learning objective is:

$$\mathcal{L} = \mathcal{L}_t + \mathcal{L}_v \tag{19}$$

## 4 EXPERIMENTS

### 4.1 Datasets

We select three typical MSA benchmark datasets: CMU-MOSI [78], CMU-MOSEI [79] and MELD [53]. The detailed partition of the dataset is included in the supplementary materials.

### 4.2 Implementation Detail

We employ text pre-trained language model Electra [19] and visual pre-trained model VGG Face2 [9], extracting features from both textual content and video frames. We use a multilayer perceptron to unify the multimodal feature dimensions and a 1-layer Transformer encoder [21] to model the multimodal data of the sequence. The batch size and epoch are set to 16 and 200, and the learning rate is configured to 7e-5. Warm up epoch is 3. Our implementation is executed using the PyTorch framework with Adam optimizer [35] on the V100 GPU.

### 4.3 Baselines

We select the state-of-the-art model in the field of MSA, MLLM and DG (OOD) as the Baseline.

**MULT** [65] designs a Multimodal Transformer to align multi-modal sequential data and capture cross-modal information interaction.

**ALMT** [80] employs non-verbal modalities to reinforce the features of the text modality several times and dismisses the non-verbal information after completing the reinforcement process.

**MAD** [54] designs a two-stage learning strategy, learning domain-specific and domain-invariant features respectively to constrain the two features by regular terms.

**RIDG** [17] aligns the labels of each class with the classified evidence to ensure the domain generalization of the model.

**MLLM**. We selected five multimodal large language models, including Blip-2 [39], InstructBlip [20], [42] and Qwen-VL [4] with excellent performance from the benchmark [75] of the multimodal large model as the baseline.

### 4.4 Evaluation Criteria

The distribution of the dataset is approximately balanced. We evaluate the model performance using a 3-class accuracy metric, specifically [Positive, Neutral, Negative].

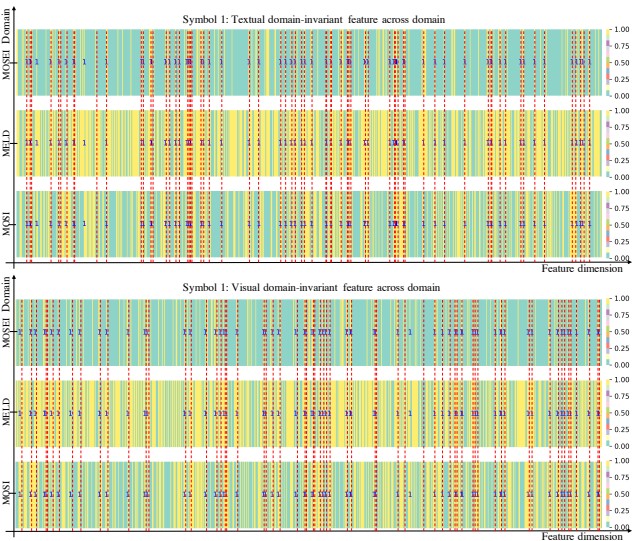

**Figure 4: Visualization of domain-invariant features across domain.**

### 4.5 Results and Discussions

*Overall Comparisons.* To justify the effectiveness of our proposed **S²LIF** model, we compareed the model with the following state-of-the-art baseline in the filed of MSA and DG. Models that focus on capturing cross-modal dependencies, called MulT and ALMT. Models that aims to learn domain-invariant features, namely, MAD and RIDG. Tables 1 and 2 show the results of the comparison. By analyzing these two tables, we draw the following conclusions: **i)** The MSA method shows visual performance in the unseen domain. With the addition of our multimodal learnable masks, the traditional models also gain the ability of DG. The fact demonstrates the effectiveness of sparse mask in DG. **ii)** Our model significantly outperforms the multimodal large model in 4 of the 6 Settings. We speculate that there is contamination from emotional datasets during the training phase of the multimodal large language model. In the two settings with better performance, the logits of InstructBlip for correctly predicted samples exceed 0.93, significantly higher than the logits generated by other multimodal large language models, which are around 0.65. **iii)** The model performance in sequential multimodal learning is better than that in non-sequential multimodal learning when we distinguish text and visual modalities. This demonstrates the effectiveness of the sequential multimodal learning strategy.

*Existence of Domain-invariant Features.* An essential assumption in our study is the presence of domain-invariant features in cross-domain multimodal data. To gain insight into this assumption, we visualized the selected and removed features for each domain using a heatmap. We marked positions with **'1'** where the features are consistently selected across domains. From Figure 4, we could conclude that there is a presence of domain-invariant features across multiple domains, and our proposed model can automatically select the domain-invariant features. Moreover, we visualized the proportion of features retained during the training phase. Figure 5

**Table 1: The performance (accuracy of 3-classification) of single-source domain generalization. The symbols $V$, $T$, and $M$ denote using visual, textual, and multimodal features, respectively. The symbols $T \rightarrow V$ and $V \rightarrow T$ indicate the multimodal learning order. $T\&V$ denotes simultaneous learning. 'Frozen' and 'Fine tuning' represents freezing and fine-tuning the parameter of pre-trained language model. We train the model on the source domain and infer on both the source and target domains.**

| Category | Method | Single-source Setting A | | | Single-source Setting B | | | Single-source Setting C | | |
|---|---|---|---|---|---|---|---|---|---|---|
| | | Source Domain | Target Domain | | Source Domain | Target Domain | | Source Domain | Target Domain | |
| | | MOSEI | MOSI | MELD | MOSI | MELD | MOSEI | MELD | MOSI | MOSEI |
| MSA | MuLT (M-Frozen) (ACL2019) | 0.644 | 0.693 | 0.516 | 0.609 | 0.344 | 0.452 | 0.663 | 0.258 | 0.435 |
| | MuLT (M-Fine tuning) (ACL2019) | 0.691 | 0.740 | 0.526 | 0.736 | 0.400 | 0.506 | 0.687 | 0.453 | 0.493 |
| | ALMT (M-Frozen) (EMNLP2023) | 0.611 | 0.688 | 0.468 | 0.548 | 0.373 | 0.465 | 0.686 | 0.306 | 0.457 |
| | ALMT (M-Fine tuning)(EMNLP2023) | 0.676 | 0.675 | 0.540 | 0.760 | 0.522 | 0.513 | 0.679 | 0.478 | 0.440 |
| MSA+Mask | MuLT + Mask (M-Frozen) (ACL2019) | 0.649 | 0.710 | 0.545 | 0.625 | 0.435 | 0.487 | 0.667 | 0.325 | 0.456 |
| | MuLT + Mask (M-Fine tuning) (ACL2019) | 0.693 | 0.759 | 0.554 | 0.766 | 0.500 | 0.553 | 0.706 | 0.519 | 0.510 |
| | ALMT + Mask (M-Frozen) (EMNLP2023) | 0.642 | 0.693 | 0.509 | 0.574 | 0.472 | 0.473 | 0.697 | 0.376 | 0.460 |
| | ALMT + Mask (M-Fine tuning) (EMNLP2023) | 0.687 | 0.749 | 0.562 | **0.777** | 0.541 | 0.586 | **0.700** | 0.553 | 0.497 |
| OOD | MAD (V) (CVPR2023) | 0.450 | 0.279 | 0.291 | 0.390 | 0.214 | 0.312 | 0.468 | 0.218 | 0.326 |
| | MAD (T-Frozen) (CVPR2023) | 0.413 | 0.172 | 0.349 | 0.344 | 0.205 | 0.334 | 0.482 | 0.154 | 0.346 |
| | MAD (T-Fine tuning)(CVPR2023) | 0.464 | 0.154 | 0.331 | 0.491 | 0.204 | 0.296 | 0.660 | 0.157 | 0.311 |
| | MAD (M-Frozen) (CVPR2023) | 0.448 | 0.304 | 0.423 | 0.374 | 0.303 | 0.346 | 0.495 | 0.243 | 0.368 |
| | MAD (M-finetune) (CVPR2023) | 0.670 | 0.744 | 0.527 | 0.746 | 0.505 | 0.563 | 0.683 | 0.419 | 0.484 |
| | RIDG (V) (ICCV2023) | 0.317 | 0.313 | 0.419 | 0.437 | 0.221 | 0.335 | 0.471 | 0.262 | 0.358 |
| | RIDG (T-Frozen) (ICCV2023) | 0.555 | 0.384 | 0.477 | 0.481 | 0.256 | 0.351 | 0.491 | 0.311 | 0.367 |
| | RIDG (T-Fine tuning) (ICCV2023) | 0.665 | 0.728 | 0.489 | 0.644 | 0.527 | 0.487 | 0.657 | 0.495 | 0.417 |
| | RIDG (M-Frozen) (ICCV2023) | 0.572 | 0.467 | 0.520 | 0.505 | 0.318 | 0.392 | 0.657 | 0.319 | 0.425 |
| | RIDG (M-Fine tuning) (ICCV2023) | 0.657 | 0.736 | 0.523 | 0.695 | 0.540 | 0.583 | 0.680 | 0.513 | 0.501 |
| MLLM | Blip-2 (ICML2023) | 0.397 | 0.290 | 0.448 | 0.290 | 0.448 | 0.397 | 0.448 | 0.290 | 0.397 |
| | InstructBlip (NeurIPS2024) | 0.540 | 0.739 | 0.492 | 0.739 | 0.492 | 0.540 | 0.492 | **0.739** | **0.540** |
| | LLava-1.5-7B (NeurIPS2023) | 0.510 | 0.351 | 0.527 | 0.351 | 0.527 | 0.510 | 0.527 | 0.351 | 0.510 |
| | LLava-1.5-13B (NeurIPS2023) | 0.453 | 0.192 | 0.496 | 0.192 | 0.496 | 0.453 | 0.496 | 0.192 | 0.453 |
| | Qwen-VL | 0.455 | 0.250 | 0.536 | 0.250 | 0.536 | 0.455 | 0.536 | 0.250 | 0.455 |
| Ours | $\mathbf{S^2LIF}$ V→T (M-Frozen) | 0.653 | 0.718 | 0.513 | 0.631 | 0.421 | 0.492 | 0.645 | 0.383 | 0.445 |
| | $\mathbf{S^2LIF}$ T&V (M-Frozen) | 0.651 | 0.720 | 0.535 | 0.629 | 0.450 | 0.504 | 0.658 | 0.380 | 0.464 |
| | $\mathbf{S^2LIF}$ T→V (M-Frozen) | 0.660 | 0.745 | 0.543 | 0.638 | 0.465 | 0.517 | 0.643 | 0.408 | 0.488 |
| | $\mathbf{S^2LIF}$ V→T (M-Fine tuning) | 0.688 | 0.759 | 0.539 | 0.775 | 0.498 | 0.606 | 0.683 | 0.529 | 0.491 |
| | $\mathbf{S^2LIF}$ T&V (M-Fine tuning) | 0.689 | 0.755 | 0.540 | 0.742 | 0.524 | 0.584 | 0.677 | 0.481 | 0.504 |
| | $\mathbf{S^2LIF}$ T→V (M-Fine tuning) | **0.701** | **0.774** | **0.572** | 0.762 | **0.556** | **0.613** | 0.692 | 0.580 | 0.519 |

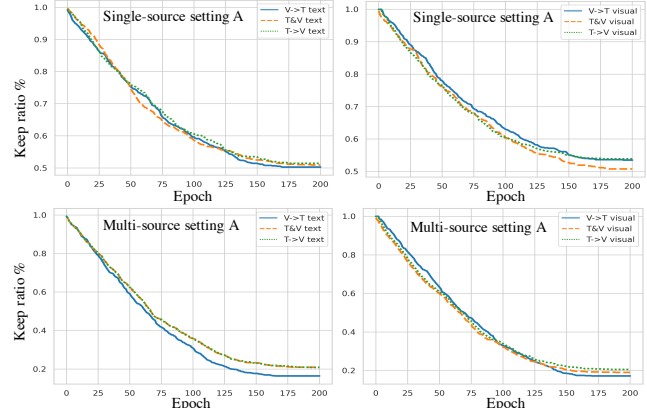

**Figure 5: The proportion of domain-invariant features.**

illustrates the proportion of features retained for both the text and visual modalities during the training phase.

*Cross-modal Feature Correlation Analysis.* Apart from the the superior performance, the key advantage of our proposed model compared to other models is that its sequential multimodal learning.

It can conditionally assist visual modalities in selecting domain-invariant features based on the domain-invariant features learned from the text modality. The features of these visual modalities prefer mutually independent from the features of the text modality, allowing the information learned from the visual modalities to complement that of the text modality. For each domain-invariant feature $x_{t_i}^{C'} \subset x_t^{c'}$ from the text modality, we employed Fisher's z-test to calculate the ratio of features in the domain-invariant feature set $x_v^{c'}$ of the visual modality that are independent and dependent of that specific feature $x_{t_i}^{c'}$. From Figure 6, we could see that, conditioning on the text modality, the model exhibits a higher proportion of independence among domain-invariant features across modalities. These results demonstrate that our proposed sequential multimodal learning strategy in Equation (4) is capable of learning more effective, sparse, and independent cross-modal features.

*Intra-modal Feature Correlation Analysis.* To valid the independence among the learned domain-invariant features, we conducted Fisher's z-test on the features in the Multi-source setting A with intra-modality. Specially, we selected $x_j^{c'}$ from the domain-invariant feature set $x^{c'}$ and computed the ratio of features in the set that are independent and dependent of $x_j^{c'}$. From Figure 7, we could be observed that, for the learned domain-invariant feature set, the

**Table 2: The performance (accuracy of 3-classification) of multi-source domain generalization.**

| Category | Method | Multi-source Setting A | | Multi-source Setting B | | Multi-source Setting C | |
|---|---|---|---|---|---|---|---|
| | | Source Domain | Target Domain | Source Domain | Target Domain | Source Domain | Target Domain |
| | | MOSEI/MELD | MOSI | MOSI/MELD | MOSEI | MOSI/MOSEI | MELD |
| MSA | MuLT (M-Frozen) (ACL2019) | 0.619/0.625 | 0.621 | 0.666/0.646 | 0.470 | 0.676/0.636 | 0.464 |
| | MuLT (M-Finetune) (ACL2019) | 0.674/0.710 | 0.660 | 0.742/0.673 | 0.549 | **0.797**/0.661 | 0.494 |
| | ALMT (M-Frozen) (EMNLP2023) | 0.630/0.669 | 0.597 | 0.660/0.685 | 0.454 | 0.664/0.591 | 0.471 |
| | ALMT (M-Finetune) (EMNLP2023) | 0.683/0.697 | 0.654 | 0.753/**0.770** | 0.528 | 0.746/0.680 | 0.516 |
| MSA+Mask | MuLT (M-Frozen) (ACL2020) | 0.647/0.659 | 0.645 | 0.644/0.688 | 0.505 | 0.730/0.654 | 0.531 |
| | MuLT (M-Finetune) (ACL2020) | 0.682/0.708 | 0.683 | 0.769/0.721 | 0.576 | 0.765/0.682 | 0.561 |
| | ALMT (M-Frozen) (EMNLP2023) | 0.640/0.657 | 0.640 | 0.645/0.682 | 0.512 | 0.673/0.631 | 0.527 |
| | ALMT (M-Finetune) (EMNLP2023) | 0.684/**0.711** | 0.681 | **0.771**/0.721 | 0.570 | 0.787/0.688 | 0.566 |
| OOD | MAD (V) (CVPR2023) | 0.437/0.468 | 0.306 | 0.365/0.411 | 0.306 | 0.355/0.436 | 0.356 |
| | MAD (T-frozen) (CVPR2023) | 0.404/0.481 | 0.306 | 0.393/0.444 | 0.319 | 0.349/0.365 | 0.388 |
| | MAD (T-finetune) (CVPR2023) | 0.444/0.691 | 0.274 | 0.432/0.653 | 0.275 | 0.438/0.481 | 0.364 |
| | MAD (M-Finetune) (CVPR2023) | 0.485/0.672 | 0.297 | 0.484/0.676 | 0.305 | 0.445/0.511 | 0.383 |
| | MAD (M-Frozen) (CVPR2023) | 0.431/0.480 | 0.316 | 0.370/0.445 | 0.349 | 0.371/0.349 | 0.428 |
| | RIDG (V) (ICCV2023) | 0.410/0.332 | 0.355 | 0.339/0.199 | 0.367 | 0.154/0.411 | 0.381 |
| | RIDG (T-frozen) (ICCV2023) | 0.548/0.623 | 0.422 | 0.561/0.643 | 0.440 | 0.571/0.552 | 0.407 |
| | RIDG (T-finetune) (ICCV2023) | 0.646/0.656 | 0.635 | 0.737/0.666 | 0.556 | 0.752/0.663 | 0.499 |
| | RIDG (M-Frozen) (ICCV2023) | 0.550/0.630 | 0.486 | 0.605/0.654 | 0.465 | 0.603/0.594 | 0.445 |
| | RIDG (M-Fine tuning) (ICCV2023) | 0.659/0.678 | 0.645 | 0.747/0.674 | 0.555 | 0.766/0.672 | 0.527 |
| MLLM | Blip-2 (ICML2023) | 0.397/0.448 | 0.290 | 0.290/0.448 | 0.397 | 0.290/0.397 | 0.448 |
| | InstructBlip (NeurIPS2024) | 0.540/0.492 | **0.739** | 0.739/0.492 | 0.540 | 0.739/0.540 | 0.492 |
| | LLava-1.5-7B (NeurIPS2023) | 0.510/0.527 | 0.351 | 0.351/0.527 | 0.510 | 0.351/0.510 | 0.527 |
| | LLava-1.5-13B (NeurIPS2023) | 0.453/0.496 | 0.192 | 0.192/0.496 | 0.453 | 0.192/0.453 | 0.496 |
| | Qwen-VL | 0.455/0.536 | 0.250 | 0.536/0.455 | 0.250 | 0.250/0.455 | 0.536 |
| **Ours** | $S^2$**LIF** $V \rightarrow T$ (M-Frozen) | 0.660/0.696 | 0.658 | 0.626/0.678 | 0.484 | 0.740/0.655 | 0.493 |
| | $S^2$**LIF** $V$ & $T$ (M-Frozen) | 0.657/0.700 | 0.659 | 0.653/0.676 | 0.487 | 0.702/0.649 | 0.505 |
| | $S^2$**LIF** $T \rightarrow V$ (M-Frozen) | 0.650/0.699 | 0.674 | 0.625/0.679 | 0.529 | 0.737/0.638 | 0.532 |
| | $S^2$**LIF** $V \rightarrow T$ (M-Finetine) | 0.679/0.712 | 0.677 | 0.758/0.707 | 0.566 | 0.778/**0.691** | 0.557 |
| | $S^2$**LIF** $V$ & $T$ (M-Finetune) | 0.686/0.706 | 0.686 | 0.762/0.704 | 0.539 | 0.768/0.671 | 0.546 |
| | $S^2$**LIF** $T \rightarrow V$ (M-Finetune) | **0.687**/0.710 | 0.687 | 0.759/0.723 | **0.581** | 0.791/ 0.687 | **0.578** |

proportion of features that are independent of any other feature in the set is significantly higher than the proportion of features that are dependent. This observation substantiates our assumption in Equation (4) that the combination of the learnable mask and classifier can effectively learn sparse and independent features.

*Correlation Analysis Between Features and Label.* To demonstrate the effectiveness of sequential multimodal learning, we also employed Fisher's z-test to analyze the correlation between the learned domain-invariant features and labels. From Figure 8, we could observe that sequential multimodal learning is capable of capturing more features that are dependent with labels. The removed features exhibit independent with the labels. This experimental result validates the efficacy of sparse masks for feature selection and the effectiveness of sequential multimodal learning (as described in Equation (3) and (4)).

*Ablation Studies.* To gain the insights into our sequential multi-modal learning strategy, We compare our model with the following variants: 1) Reordering sequence learning, including $T \rightarrow V$, $V \rightarrow T$, $T$&$V$, where they respectively denote sequential multimodal learning with textual modality as the condition, with visual modality as the condition, and simultaneous learning of textual and visual modality. 2) **Add Noise**, introducing noise by replacing domain-invariant features with random noise as evidence for the classifier. 3) **Using DS**, utilizing domain-specific features as evidence for the classifier. 4) **w/o key-frame mask**, eliminating the

**Table 3: Ablation study on Multi-source Setting A.**

| Method | Multi-source Setting A | |
|---|---|---|
| | Source Domain | Target Domain |
| | MOSEI/MELD | MOSI |
| Add Noise | 0.367/0.199 | 0.339 |
| Using DS | 0.372/0.202 | 0.341 |
| w/o key-frame mask | 0.642/0.695 | 663 |
| $S^2$**LIF** $T \rightarrow V$ (M-Frozen) | **0.650/0.699** | **0.674** |

key-frame masking module. From Table 1, 2 and 3, we could see that leveraging the text modality as a condition yields higher performance. Table 2 reveals that replacing the learned domain-invariant features with noise results in a modest performance decline, with domain-specific features outperforming random noise to a slight extent. These observations reflect the following key insights: 1) The effectiveness of sequential multimodal learning. 2) The capability of our model to efficiently learn domain-invariant features. 3) Our model effectively eliminates domain-specific features that do not contribute significantly to classification.

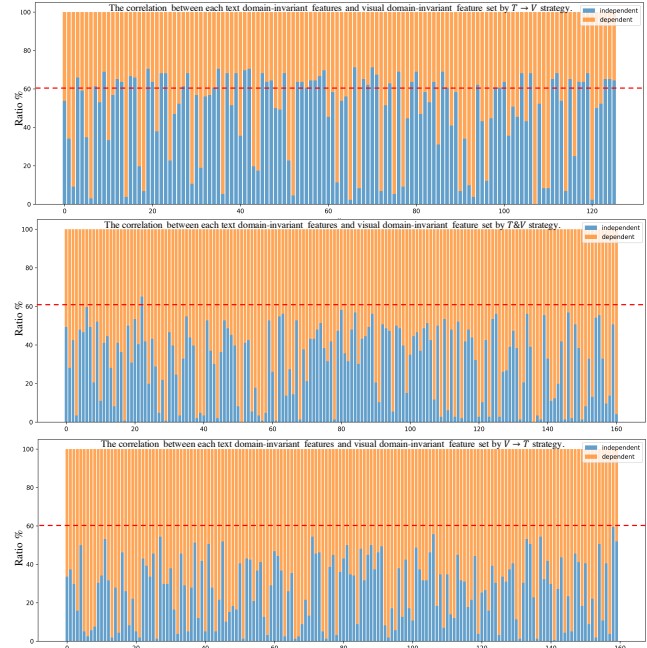

Figure 6: X-axis: Single textual domain-invariant feature. Y-axis: The independent and dependent ratio of the visual domain-invariant feature set to the each textual domain-invariant feature.

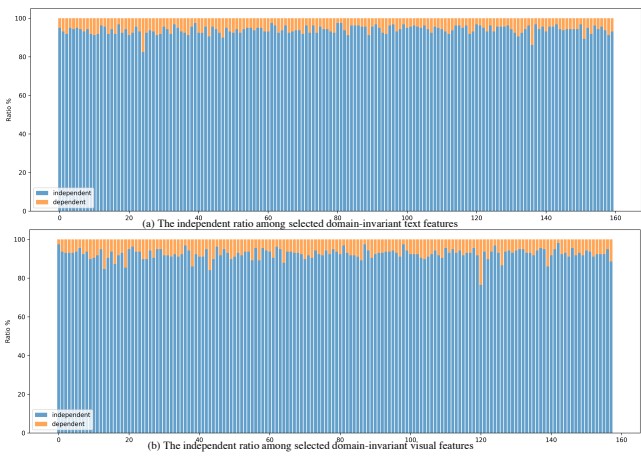

Figure 7: X-axis: Single domain-invariant feature of intra-modality. Y-axis: The independent and dependent ratio between single domain-invariant feature and the other domain-invariant features of intra-modality.

### 4.6 Case Study.

To qualitatively validate the effectiveness of our proposed model, we showcase the predictive outcomes of our model on several samples, encompassing positive, negative, and neutral sentiments. As shown in Figure 9, our model demonstrates accurate recognition of all

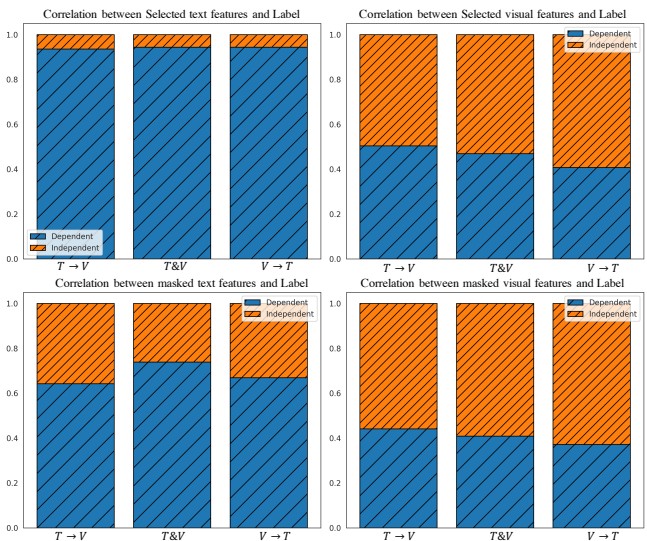

Figure 8: The correlated proportion of domain-invariant and domain-specific features with label.

| Modal Content | Ground Truth | Prediction |
|---|---|---|
| This movie isnt just *bad* its diabolical. | Negative | Negative ✓ |
| And he really seems to be channeling michael bay in this movie. | Neutral | Neutral ✓ |
| But he has some of the *funniest* scenes in this movie. | Positive | Positive ✓ |

Figure 9: The predictions on the testset of Multi-source Setting A.

three sentiment polarity. This indicates the robustness of our model on unseen domains.

## 5 CONCLUSION

In this paper, we design a sequential multimodal learning strategy to learn cross-domain invariant features for MSA. Specifically, we first employ learnable masks and classifiers to learn the invariant features from texts, and then select the invariant features of videos, conditioned on the selected text features. The experiment demonstrates the efficacy of our model in both single-domain and multi-source domain settings. Based on extensive experiments, we conclude that i) the learning order between modalities is important for domain generalization performance, and ii) our learning strategy prefers the selection of features that are statistically independent to each other, in particular between modalities.

In the future, we will consider including more modalities, such as audio modality, to analyze the correlation between cross-modal invariant features in cross-domain scenarios.

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
