# OpenReview forum: "Learning in Order!   A Sequential Strategy to Learn Invariant Features for Multimodal Sentiment Analysis"
_acmmm.org/ACMMM/2024/Conference — MM2024 Poster_

### Official Review · Reviewer_6AUc · 2024-05-21

**Rating:** 2
**Confidence:** 4

**Summary:**

The paper proposes a novel and simple sequential learning strategy to train models on videos and texts for multimodal sentiment analysis. This strategy starts with learning domain invariant features from texts, followed by learning sparse domain-agnostic features from videos, assisted by the selected features learned in texts. This paper introduces a multimodal model that is trained either in a single source domain or multiple source domains using our learning strategy. The experimental results demonstrate that the multimodal model achieves significantly better performance than the state-of-the-art approaches on average in both single-source and multi-source settings.

**Strengths:**

The advantage of this article is that it proposes a learning strategy to learn domain-invariant text features and visual features separately to improve domain generalization. The idea is simple and intuitive.  Experimental results verify the performance of the method.

**Limitations:**

This idea is more like a specific strategy, and some loss functions are also designed to improve performance. The key idea of this idea is to first learn the invariant features from texts, and then select the invariant features of videos, conditioned on the selected text features. It emphasizes that the learning order between modalities is important (Title: Learning in Order! A Sequential Strategy to Learn Invariant Features for Multimodal Sentiment Analysis) and learning all modalities simultaneously leads to inferior performance, which is very confusing and has no experimental evidence. The article states in the conclusion that the proposed learning strategy tends to select statistically independent features, especially between modalities, and there is no reasonable explanation for this. Therefore, the plausibility of this article is unclear.

Detailed comments:

1.	In Section 3 (Method), the role of each learning objective is not well explained.
2.	The various strategies in Section 3 (Method) are a bit messy, and the reading does not clearly reflect which strategy improves which part of the model performance.

**Suitability:**

2

---

### Official Review · Reviewer_ehDt · 2024-05-26

**Rating:** 5
**Confidence:** 1

**Summary:**

This work proposes a novel and simple sequential learning strategy to train models on videos and texts for multimodal sentiment analysis.

**Strengths:**

I'm very sorry, but I don't have enough time to review your article in detail. I think the article is well written, so I give it a higher rating.

**Limitations:**

I'm very sorry, but I don't have enough time to review your article in detail. I think the article is well written, so I give it a higher rating.

**Suitability:**

3

---

### Official Review · Reviewer_PRFQ · 2024-06-01

**Rating:** 5
**Confidence:** 3

**Summary:**

The paper proposes a sequential learning strategy to train models on video and texts for multimodal sentiment analysis.

**Strengths:**

The article is clear and well written.

**Limitations:**

The concept of multimodal sentiment analysis leveraging sequential transfer learning is well-established in the field, and numerous studies have already explored this approach. Given this context, the primary shortcoming of the presented paper lies in its lack of novelty. While the paper might offer some incremental improvements or variations on existing methods, it does not introduce a fundamentally new concept or approach that significantly advances the state-of-the-art in multimodal sentiment analysis. Moreover, the reliance on sequential transfer learning, while effective, has been extensively documented and utilized in prior research, diminishing the impact of its use in this context. In terms of experimental results, Table 1 shows better results for single-source settings A and single-source settings B, but not in single-source settings C where other method return best experimental results. In Table 2 in multi-source settings C there are evident better performances but in multi-source settings A and B other methods compared with the one proposed in the paper, return better results in the source domain. Overall, the experimental results demonstrate good performance, making the article acceptable despite its lack of novelty. The findings suggest that the methods employed are effective and yield satisfactory outcomes, contributing positively to the body of research in multimodal sentiment analysis. While the approach itself may not be groundbreaking, the good results indicate that the techniques used are robust and reliable. This level of performance ensures that the paper holds value and relevance within the field, justifying its acceptance.

**Suitability:**

2

---

### Official Review · Reviewer_NXR7 · 2024-06-01

**Rating:** 4
**Confidence:** 2

**Summary:**

The authors address a very interesting task with an innovative architecture. In detail, it proposes a new sequential learning strategy for multimodal sentiment analysis, training models on videos and texts, with better performance than state-of-the-art methods, using domain-invariant and sparse domain-agnostic features.

**Strengths:**

The proposed architecture seems innovative, applies special ‘attention’ mechanisms and succeeds in achieving excellent results in a multimodal context

**Limitations:**

The structure of the paper must be improved and some details of the supplementary material must necessarily be added to the main paper. Below are some notes to improve the paper:
- To get an overview of how the paper is organised, the structure of the paper could be added at the end of the introduction section.
- The section on the method requires several changes and improvements in terms of description in order to understand well the proposed approach. For example, what ‘n’ (not ‘N’) represents on page 2, line 183 (method section). Check and improve the whole document in this respect.
- The details of the dataset and its partition information (trainset (%), validation set (%) and test set (%)) should be added in the main document!

**Suitability:**

3

---

### Official Review · Reviewer_9QYC · 2024-06-04

**Rating:** 4
**Confidence:** 3

**Summary:**

The paper titled "Learning in Order! A Sequential Strategy to Learn Invariant Features for Multimodal Sentiment Analysis" introduces a novel and straightforward sequential learning strategy for training models on videos and texts for multimodal sentiment analysis (MSA). This approach aims to estimate sentiment polarities on unseen out-of-distribution data by training a multimodal model in either a single source domain or multiple source domains. The strategy involves first learning domain-invariant features from text, followed by learning sparse domain-agnostic features from videos, guided by the features learned from text. The experimental results show that this model significantly outperforms state-of-the-art methods in both single-source and multi-source settings. Additionally, the paper highlights a feature selection procedure that favors features independent of each other and strongly correlated with their polarity labels. The authors also commit to making the source code publicly available upon acceptance of the paper.

**Strengths:**

This paper is notable for its innovative sequential learning strategy, which emphasizes the importance of the learning order between modalities for domain generalization performance. The theoretical approach is sound, as it effectively integrates domain-invariant feature learning from text with domain-agnostic feature learning from videos, leading to a more robust MSA model. The comprehensive evaluation across single-source and multi-source settings demonstrates the model's effectiveness and generalizability. The clarity of the methodology, supported by extensive experimental validation, underscores the technical correctness and practical applicability of the approach. Furthermore, the paper provides a detailed feature selection process that enhances the model's performance by focusing on statistically independent features, which is a novel contribution to the field. The commitment to open-source the code further adds to the paper's value by facilitating future research and development.

**Limitations:**

Despite the strengths, the paper has several limitations that need addressing. Firstly, the presentation of the content is somewhat difficult to follow, making it challenging for readers to grasp the core contributions and methodology. Figures 4, 6, 7, 8, and 9 are too small and nearly illegible in the printed version, which hampers the reader's ability to understand the experimental results fully. Moreover, while the bibliography is extensive, spanning two pages for a ten-page paper seems excessive and could be streamlined for better readability. There is also a typo at the beginning of paragraph 4.5 ("we compared"), which needs correction. Lastly, although the paper is interesting and makes a valuable contribution, it would benefit from a more thorough discussion of its limitations, such as potential scalability issues or the impact of different modalities not yet explored, like audio. Given these points, the article is accepted with revisions as outlined above.

**Suitability:**

3

---

### Meta-Review · Area_Chair_zyUh · 2024-07-01

**Recommendation:** Accept (Poster)
**Confidence:** 4

**Metareview:**

The paper received mixed scores. Some reviewers incresased the evaluation to BA after rebuttal as some concerns have been cleared. The average score is on the positive side, however there still are concerns.